# Discovery of Flavonoids as Novel Inhibitors of ATP Citrate Lyase: Structure–Activity Relationship and Inhibition Profiles

**DOI:** 10.3390/ijms231810747

**Published:** 2022-09-15

**Authors:** Pan Wang, Tao Hou, Fangfang Xu, Fengbin Luo, Han Zhou, Fan Liu, Xiaomin Xie, Yanfang Liu, Jixia Wang, Zhimou Guo, Xinmiao Liang

**Affiliations:** 1CAS Key Laboratory of Separation Science for Analytical Chemistry, Dalian Institute of Chemical Physics, Chinese Academy of Sciences, Dalian 116023, China; 2University of Chinese Academy of Sciences, Beijing 100049, China; 3Jiangxi Provincial Key Laboratory for Pharmacodynamic Material Basis of Traditional Chinese Medicine, Ganjiang Chinese Medicine Innovation Center, Nanchang 330100, China

**Keywords:** ACLY inhibitor, flavonoid, structure–activity relationship, herbacetin

## Abstract

ATP citrate lyase (ACLY) is a key enzyme in glucolipid metabolism and its aberrantly high expression is closely associated with various cancers, hyperlipemia and atherosclerotic cardiovascular diseases. Prospects of ACLY inhibitors as treatments of these diseases are excellent. To date, flavonoids have not been extensively reported as ACLY inhibitors. In our study, 138 flavonoids were screened and 21 of them were subjected to concentration–response curves. A remarkable structure–activity relationship (SAR) trend was found: ortho-dihydroxyphenyl and a conjugated system maintained by a pyrone ring were critical for inhibitory activity. Among these flavonoids, herbacetin had a typical structure and showed a non–aggregated state in solution and a high inhibition potency (IC_50_ = 0.50 ± 0.08 μM), and therefore was selected as a representative for the ligand–protein interaction study. In thermal shift assays, herbacetin improved the thermal stability of ACLY, suggesting a direct interaction with ACLY. Kinetic studies determined that herbacetin was a noncompetitive inhibitor of ACLY, as illustrated by molecular docking and dynamics simulation. Together, this work demonstrated flavonoids as novel and potent ACLY inhibitors with a remarkable SAR trend, which may help design high–potency ACLY inhibitors. In–depth studies of herbacetin deepened our understanding of the interactions between flavonoids and ACLY.

## 1. Introduction 

ATP citrate lyase (ACLY) catalyzes the transformation of citric acid to acetyl-CoA [1,2]. Citric acid comes from the tricarboxylic acid cycle (TCA) and acetyl-CoA is an important precursor in the biosynthesis of fatty acids and cholesterol. Thus, ACLY is a key enzyme in glucolipid metabolism by linking TCA and lipid anabolism [3]. Abnormally high expression of ACLY is associated with multiple diseases including cancers [4,5,6,7,8,9], dyslipidemia [10,11] and atherosclerosis [12], making ACLY an attractive pharmaceutical target. ACLY inhibitors have exhibited prominent prospects as treatments for these diseases. For example, Nexletol (bempedoic acid or ETC-1002), the first drug to target ACLY, was approved in 2020 for the clinical treatment of hyperlipemia and atherosclerotic cardiovascular disease because of its significant LDL-C lowering effect [13,14]. However, Nexletol is a prodrug that needs to be activated by acyl-CoA synthetase long–chain family member 1 (ACSVL1) to exert ACLY inhibition and the therapeutic effects of Nexletol [12]. Therefore, new ACLY inhibitors may provide opportunities for drug development.

Hydroxycitric acid (HCA) is the first reported ACLY inhibitor with a *K_i_* value of 3 μM [15] and is used as a popular natural supplement and a promising agent for obesity. However, due to its low cell penetration, high concentrations of HCA will be needed to totally inhibit ACLY [7]. Long–chain dioic acids known as Medica analogues were reported to be potent hypolipidemic drugs in rats and act on ACLY, but the highest potency was only around 16 μM [16]. Later, 2-substituted butanedioic acids were designed against ACLY with K_i_ in a 1–3 μM range, but none of them lowered lipid synthesis in HepG2 cells, like SB–201076, which may be caused by the lack of cell membrane permeability resulting from a polar carboxyl group of the molecules [10]. When the carboxyl was transformed to lactone (SB–204990), a prodrug of the active inhibitor SB–201076, lipid–lowering and anticancer effects were observed both in vitro and in vivo [17,18], but no clinical practice has been reported, which may be due to the lack of tissue specificity. In 2007, 2-hydroxy-N-arylbenzenesulfonamides were identified as potent ACLY inhibitors and the lead inhibitor BMS–303141 lowered plasma cholesterol, triglyceride and glucose levels in mice fed on a high-fat diet [19]. NDI–0911143, carrying a benzenesulfonamide scaffold, inhibited human ACLY with a nanomolar potency [20], but no clinical research has been reported yet. In summary, existing ACLY inhibitors are mainly divided into two categories. One shares a terminal dicarboxylic acid in the structure, considered as analogues of citric acid, and competitively inhibits ACLY with a relatively weak potency and low membrane permeability. The other is benzenesulfonamide scaffold inhibitors, which have potent activities, but no clinical research has been conducted thus far. Currently, the structure of ACLY inhibitors is insufficient and discovery of inhibitors with structural novelty and high activity continues to attract the interest of researchers.

With high structural diversity, natural products are important resources of molecules that exhibit distinctive pharmacological or biological activities. Flavonoids are important class of natural products with anticancer, antioxidant, antiviral and anti-inflammatory effects [21,22]. For example, herbacetin suppresses melanoma cell growth [23] and morin is related to lipid peroxidation [24]. Rutin and quercetin are promising agents for the treatment of atherosclerotic cardiovascular disease [25]. Generally, these diseases are accompanied by abnormal ACLY expression. T ACLY may serve as a potential target of some flavonoids. Recently, luteolin, a common polyhydroxy flavonoid, has been reported to inhibit ACLY [26]. However, general structure–activity relationship (SAR) studies of flavonoids are still lacking. 

With the goal of identifying novel and potent ACLY inhibitors, we conducted a high-throughput screening of 138 commercially available flavonoids and selected 21 of them to learn the SAR. A cell-free aggregation assay was conducted to exclude pseudopositive results. Meanwhile, we used herbacetin as an example for an in-depth study of its interaction with ACLY by thermal shift assay, kinetic studies, and molecular docking and dynamics simulation.

## 2. Results

### 2.1. Preliminary Screening of Natural Flavonoids for ACLY Inhibitors

To identify potent and novel ACLY inhibitors, 138 commercially available natural flavonoids were screened (Appendix A). There were some flavonoids that showed considerable inhibition efficacy compared to BMS−303141, the positive ACLY inhibitor (Figure 1A). Among them, 29 flavonoids showed inhibition rates higher than 50% at the concentration of 10 µm. Interestingly, we discovered that ortho−dihydroxyphenyl flavonoids were more potent than other analogues, for example, compounds **18** (herbacetin), **29** (quercetin), **51** (luteolin), **52** (scutellarein) and **89** (myricetin) that strongly inhibited ACLY activity with inhibition rates of 96.07%, 91.09%, 88.28%, 91.80% and 93.29%, respectively (Figure 1B,C). The results indicated that flavonoids may be a potential class of ACLY inhibitors, and the importance of ortho-hydroxyphenyl warrants further study.

### 2.2. SAR Analysis of Flavonoids

We selected the Table 1 flavonoids from the 138 based onthe screening results and divided them into four subtypes. According to concentration–response curves, all these flavonoids showed inhibition in a concentration-dependent manner (Figure 2). Generally, ortho–dihydroxyphenyl flavonoids, including herbacetin, quercetin, luteolin, scutellarein, myricetin and gossypetin, showed IC_50_ values lower than 1 μM (Figure 2A and Table 1). Meta–dihydroxyphenyl flavonoids, including kaempferol, apigenin, isohamnetin, and morin, showed IC_50_ values between 4.07 μM and 49.74 μM (Figure 2B and Table 1). Flavonoid glycosides, such as hyperoside, isoquercetin, luteolin 7-O-glucuronide, vincetoxicoside B, isoorientin, scutellarin and lonicerin, had IC_50_ values ranging from 0.57 μM to 3.77 μM (Figure 2C and Table 1). The remaining four flavonoids that were different in the pyrone ring, including taxifolin, catechin, epicatechin and cyanidin, displayed IC_50_ values from 1.29 μM to 31.63 μM (Figure 2D and Table 1).

In agreement with the preliminary screening result, ortho–dihydroxyphenyl flavonoids exhibited high inhibition potency out of the 21 flavonoids. The activity of luteolin with 4′,5′-dihydroxy was 30-fold higher than that of apigenin that lacked 4′-hydroxy. Similarly, compared with kaempferol, herbacetin was 20-fold more active and quercetin was 10-fold more active. Once the 5′-hydroxy was replaced by a methoxy group (isohamnetin), the inhibition efficacy almost disappeared. Morin with meta-dihydroxy rather than ortho-dihydroxyphenyl was less active than quercetin, indicating the indispensability of ortho-dihydroxyphenyl in achieving high flavonoid activity. Interestingly, the position of ortho-dihydroxyphenyl did not matter much; for example, herbacetin and quercetin with ortho-dihydroxyphenyl at different positions shared equivalent activity. Subsequently, we investigated structures with more ortho-hydroxy groups. Myricetin with a 3′,4′,5′-trihydroxy substituent (IC_50_ = 0.57 ± 0.03 μM) and gossypetin with a 3′,4′-dihydroxy substituent and a 7,8-dihydroxy substituent (IC_50_ = 0.31 ± 0.02 μM) were not more active than herbacetin and quercetin, respectively.

Next, we investigated the glycoside substitution at different positions. Compared to luteolin and quercetin, the introduction of 3-glycoside (hyperoside and isoquercetin, respectively) mildly reduced the activity, while a 7-position substitution had little effect, whether the group was glucose (vincetoxicoside B) or disaccharide (lonicerin). However, luteolin 7-O-glucuronide with replacement of a gluconic acid was less potent. Additionally, the introduction of 6-glycoside substitution (isoorientin) lowered the activity.

Regarding the remaining four flavonoids different in C ring (Figure 2D and Table 1), we found that a carbonyl group and a double bond of pyrone favored the activity. When quercetin was modified to a flavononol(taxifolin), its activity decreased, as did the carbonyl group (cyanidin). If both carbonyl and double bond were absent, the potency would decline greatly (catechin and epicatechin). These results suggested that a conjugated system on the pyrone ring should help flavonoids maintain ACLY inhibitory activity.

### 2.3. The Aggregation Analysis of 3 Ortho–Dihydroxyphenyl Flavonoids

At some experimental conditions, pseudopositive results of flavonoids may be observed due to compound aggregation in solution [27]. Therefore, we performed aggregation assay on three high–potency flavonoids to avoid this possibility. Herbacetin induced negligible DMR responses even at 100 μM, indicating a monomer state (Figure 3A,D). Quercetin triggered low signals at concentrations below 6 μM, but positive DMRs were observed when the concentration was above 6 μM (Figure 3B,E). To determine the critical aggregation concentration (CAC) of quercetin, we collected and analyzed the DMR responses of different concentrations at a fixed time. The intersection of the horizontal line and the oblique line was its CAC value of 6.47 μM, which was 7 times higher than its IC_50_ value (0.86 ± 0.05 μM) (Figure 3E). Similar analysis was conducted on myricetin and its CAC value was 1.82 μM, 3-fold higher than its IC_50_ value (0.57 ± 0.03 μM) (Figure 3C,F). These results indicated that ACLY inhibition activity was not caused by flavonoid aggregation in the reaction buffer and that the IC_50_ values were authentic.

### 2.4. Direct Interaction of Herbacetin with ACLY

Based on the above results, we chose herbacetin as a representative flavonoid for further study. Drug–target interactions are critical for drug discovery. We performed a thermal shift assay to study interactions between herbacetin and ACLY. In the heating process, ACLY degenerated gradually, and fluorescence signals changed in the presence of Sypro Orange, generating a melting curve simultaneously. As shown in Figure 4A, the melting curves of ACLY treated with either 40 μM or 200 μM herbacetin shifted to the right compared with the control group treated with DMSO. The T_m_ values of the control, 40 μM and 200 μM herbacetin-treated groups were 54.5 °C, 55.5 °C and 56.5 °C, respectively (Figure 4B), indicating improved thermostability of ACLY by herbacetin in the heating process. Western blotting analysis also supported this finding. At a discontinuous temperature gradient, ACLY samples treated with 200 μM herbacetin were more stable, and the grayscale values of the undenatured ACLY samples were significantly higher than those of the control at 52 °C, 57 °C, 60 °C, and 62 °C (Figure 4C,E). At 57 °C, stability of ACLY treated with herbacetin at different concentrations showed a concentration–dependent effect (Figure 4D,F). These results demonstrated that herbacetin directly bound ACLY and enhanced its thermal stability.

### 2.5. Herbacetin Was a Noncompetitive Inhibitor of ACLY

Enzymatic kinetic studies were conducted to investigate the binding site and inhibition pattern of herbacetin on ACLY, using citric acid (CA) as substrate. Lineweaver-Burk curves converged at (−1/*K_m_*, 0) (Figure 5A), indicating that herbacetin was a noncompetitive inhibitor towards CA. *K_m_* was calculated to be 47.74 μM and *K_i_* was calculated to be 0.11 μM. Similarly, herbacetin was identified as a noncompetitive inhibitor of ATP with *K_m_* = 82.73 μM and *K_i_* = 0.29 μM (Figure 5B), and a noncompetitive inhibitor of CoA with *K_m_* = 77.32 μM and *K_i_* = 0.19 μM (Figure 5C). Together, herbacetin was characterized as a noncompetitive inhibitor of ACLY.

### 2.6. Docking Studies

NDI–091143 was a reported allosteric ACLY inhibitor and its binding domain was illustrated in the co–crystallized complex (PDB: 6O0H). Based on the kinetic studies, we deduced that herbacetin interacted with ACLY at a site different from the three substrates. There was a possibility that herbacetin bound at the allosteric site as NDI–091143 did. To validate our hypothesis, we performed a docking study. 

The docking pocket was set as a grid of 10 Å × 10 Å × 10 Å. Before docking, the original ligand NDI–091143 was re-docked into the pocket to validate the docking procedure. The output conformation after docking and the original conformation were almost superimposed with a room mean square deviation (RMSD) value of 0.29 Å (Appendix A), indicating that the docking procedure had good reproducibility for receptor–ligand complexes. Then we docked herbacetin into this preared pocket of ACLY after removal of NDI–091143 and analyzed the interactions between herbacetin and ACLY. Figure 6 showed that herbacetin had hydrogen-bond interactions with ASN346, THR348, ASN349 and THR353. The benzene ring of benzopyrone formed a π–π stacking with PHE354. The main amino acids responsible for NDI–091143 were PHE347, PHE354, ARG378 and GLY380. The results were in consistent with the kinetic studies.

### 2.7. Molecular Dynamics Simulation

To evaluate the stability of the interactions between ligands and ACLY, the docked complex was conducted under molecular dynamics simulation of 200 ns in Desmond module. RMSD measures the degree to which the overall conformation of the protein and ligand changes during the simulation. The left *y*-axis represented the RMSD trend of the protein, and the right *y*-axis represented the stability of the ligand with respect to the protein and its binding pocket.

When herbacetin was bound to ACLY, the RMSD of the protein conformation fluctuated obviously from 0 to 150 ns and was relatively stable after 150 ns (Figure 7A). When NDI–0911143 was bound, the timepoint was 55 ns (Figure 7B). During the 200 ns simulation, neither herbacetin nor NDI–091143 changed posture or position notably. Root mean square fluctuation (RMSF) represented the positional changes of each residue in the protein over the simulation time period. The amino acids of C–terminal regions fluctuated more than the rest of the residues (Figure 7C,D), while the key amino acids illustrated by Appendix A fluctuated with a small RMSF, as expected (from 2 Å to 5 Å). In the interactions between ACLY and herbacetin, ARG378, THR353, ASN349, ILE344 and SER343 formed hydrogen bonds; ILE344, PHE354 and ILE340 exhibited hydrophobic interactions; GLU669 formed water bridges with surroundings (Appendix A). In the interactions between ACLY and NDI–091143, ARG378, GLY380, GLY381 and GLY342 formed hydrogen bonds; TYR307, ILE340, PHE347, PHE354, ILE357 and VAL377 exhibited hydrophobic interactions; ARG379 and PRO382 formed water bridges with surroundings (Appendix A). Appendix A documented the interactions between ACLY and herbacetin (Appendix A) or NDI–091143 (Appendix A) throughout the MD simulation. The key amino acids that appeared were consistent with the above results. Looking into the results of MD, the conformational changes of ACLY and the key amino acids exhibited in the binding of herbacetin or NDI–091143 varied greatly, and this might help us understand why the inhibition pattern of herbacetin was different from that of NDI–091143.

## 3. Discussion

ACLY is considered a potential target for various diseases, such as cancer, dyslipidemia, and cardiovascular diseases [2,3,4,8,12] and thus, the development of ACLY inhibitors becomes important. In this work, we screened numerous flavonoids and found multiple highly active ACLY inhibitors, including herbacetin, luteolin, quercetin, and gossypetin. These flavonoids, except luteolin, were reported to be ACLY inhibitors for the first time. Flavonoids as ACLY inhibitors were structurally different from the reported inhibitors, such as the well-known Medica–16 [16], clinically used Nexletol [13], and SB–204990 [17], which share a terminal carboxyl group, and were also different from the more active 2-hydroxy-N-arylbenzenesulfonamides inhibitors BMS–303141 [19] and NDI–091143 [20]. Additionally, as natural ACLY inhibitors, seven flavonoids had IC_50_ values lower than 1 μM, and were markedly more active than hydroxycitric acid (*K_i_* = 300 μM), antimycin (*K_i_* = 4 μM), purporone (IC_50_ = 8 μM), and radicicol (*K_i_* = 10 μM), and equivalent to anthrone (IC_50_ = 280 nM) and isochlorogenic acid A (IC_50_ = 0.10 μM), which, to the best of our knowledge, is the most potent natural ACLY inhibitor. Our findings expanded the category of ACLY inhibitors, and these highly active flavonoids are promising candidates for pharmacological and mechanistic research. In addition, since ortho-dihydroxyphenyl is critical for inhibition activities, we speculate that the hydroxyphenyl group has non-negligible interactions with ACLY residues. It was reported that catecholic group at the B ring or A ring tended to form orthoquinones and covalently bind on biothiols or the cysteines in target proteins [28]. However, we found that in the interactions with ACLY, hydroxyphenyl more likely served as a hydrogen bond donor to exert its function rather than the form of orthoquinones (Appendix A). What is more, one of the phenolic hydroxyl groups might change the angle of the molecule to facilitate interactions with ACLY residues. However, this hypothesis warrants further studies.

We noted a clear SAR trend in which the ortho-dihydroxyphenyl and the conjugated system on the pyrone ring were key structural factors of flavonoids for ACLY inhibition. The hydroxyphenyl structure, especially ortho-dihydroxyphenyl in catechol, was easily oxidized into a quinone structure and not good for druggability. To make more stable structures, one hydroxyphenyl could be replaced by appropriate groups. We observed that methoxy or isopentene substitution would lead to poor activity (isorhamnetin), while halogens such as chlorine or bromine might be suitable, as suggested by the SAR of emodin derivatives [29].

Computational methods have emerged as a fast and inexpensive approach for studying the interactions of ligands and proteins [30,31]. The MM–GBSA method has been successfully developed to predict the free energy of binding between small molecules and targets. Different docking software adopted different algorithms. In our article, we used the prime module of Schrödinger software to perform MM–GBSA calculations. The binding affinity of protein–ligand complex crystal structures can be studied in terms of free binding energy (ΔG_bind_). ΔG_bind_ mainly has three influence factors: gas phase free energy (ΔG_MM_), solvation free energy (ΔG_solv_) and system entropy change (ΔG_SA_). The relationship between the three is as follows: ΔG_bind_ = ΔG_MM_ + ΔG_solv_ + ΔG_SA_. We selected the solvent model as VSGB, set the ligand and receptor amino acid position, orientation, and conformation, and performed hierarchical sampling. As a result, the MM–GBSA values of NDI–091143 and herbacetin were −57.26 and −50.94, respectively, indicating that there was little difference between them. This means that the herbacetin is stable in the ACLY binding pocket. Additionally, we docked another five potent flavonoids (quercetin, luteolin, scutellarein, myricetin and gossypetin) and performed MM–GBSA calculation (Appendix A). Their MM–GBSA calculation values were greater than that of NDI–091143 and were consistent with the activity data. The results also supported the applicability of MM–GBSA in virtual screening.

Herbacetin has several medicinal properties including anticancer, antioxidant, anti-inflammatory and antibacterial effects [32,33,34]. AKT and ornithine decarboxylase were reported to be its target. Herein, we defined ACLY as herbacetin’s direct target. These targets had a certain correlation in terms of indications, and it could be speculated that they act synergistically. Herbacetin could exert its efficacy through multiple targets. It was meaningful to study the mechanism of synergy at a molecular level.

## 4. Materials and Methods

### 4.1. Reagents and Materials

Recombinant His-tagged ACLY was purified by a Ni-NTA affinity chromatography column. The plasmid, gifted by Professor Liang Tong (Columbia University, New York, NY, USA), was transformed into *Escherichia coli* BL21(DE3) and cultured on LB solid medium, and then the bacteria were scaled up in culture flasks after single clones were picked. Then isopropyl-beta-D-thiogalactopyranoside was added to induce the expression of recombinant ACLY. Bacteria were lysed and affinity purification was conducted to obtain high purity ACLY. ADP-Glo™ Kinase Assay Kit was purchased from Promega Corporation (Madison, WI, USA). BMS–303141 was bought from MedchemExpress (Princeton, NJ, USA). Commercially available natural flavonoids were obtained from Shanghai Yuanye Biotechnology Co., Ltd. (Shanghai, China). Sypro Orange was from Thermo Fisher Scientific (Shanghai, China). Low-profile PCR tubes and optical flat 8-cap strips were purchased from Bio-Rad company (Hercules, CA, USA). Cell counting kit-8, RIPA lysis buffer (strong), BCA protein assay kit, SDS–PAGE sample loading buffer (5×), SDS–PAGE electrophoresis buffer (Tris-Gly, 10×), BeyoECL Star and PMSF (100 mM) were from Beyotime Biotechnology (Shanghai, China). Anti-ATP citrate lyase was bought from Abcam (Cambridge, UK). Anti-mouse IgG was purchased from Cell Signaling (Danvers, MA, USA). A549 cells and PC3 cells were from the Type Culture Collection of the Chinese Academy of Science (Shanghai, China). 

### 4.2. Enzymatic Inhibition Assay

We established a robust enzymatic assay based on ADP-Glo™ Kinase Assay. The reaction buffer was prepared by tris 40 mM (pH 8.0), dithiothreitol 4 mM, MgCl_2_ 10 mM, dimethyl sulfoxide 1% (DMSO, *w*/*v*), Brij35 0.01% (*w*/*v*), and bovine serum albumin 0.001% (*w*/*v*). First, purified ACLY was preincubated with compounds of different concentrations for 15 min. Next, a substrate mixture containing citric acid (CA), ATP and CoA was added to start the reaction at 37 °C for 60 min. The final concentrations of ACLY, CA, ATP and CoA were fixed at 7.5 nM, 600 μM, 150 μM and 150 μM, respectively. Then ADP-Glo™ Reagent was added and incubated for 40 min at room temperature to stop the reaction and consume excessive ATP. Finally, Kinase Detection Reagent converted ADP to ATP, which was utilized by firefly luciferase to generate bioluminescence detected by a microplate reader (EnSight, PE, Waltham, MA, USA).

### 4.3. Cell–Free Aggregation Assay

The aggregation assay was performed using an Epic BT system (Corning, NY, USA). Reaction buffer (30 μL) was aliquoted in an Epic 384-well microplate (Corning, NY, USA) and equilibrated for 1 h. Then 10 μL compound solution was added and the microplate was placed on the Epic reader. Dynamic mass redistribution (DMR) signals were recorded for 1 h.

### 4.4. Thermal Shift Assay

The thermal shift assay was conducted in a real-time PCR machine (Bio-Rad, Hercules, CA, USA). Sypro Orange fluorescent dye was used to monitor protein folding/unfolding. Purified ACLY and compound solutions were preincubated in low-profile PCR tubes for 15 min at room temperature, and then Sypro Orange was added and incubated for another 15 min on ice. The total volume was 20 μL. The final concentration of ACLY was 1.45 μM and Sypro Orange was 12.5×. The PCR tubes were heated from 25 to 90 °C at a heating rate of 0.5 °C/10 s. The fluorescence intensity changes in each well were recorded. T_m_ (transition midpoint) was read from the melting curve.

### 4.5. Western Blotting Analysis

Purified ACLY was preincubated with compounds or DMSO in tubes at room temperature for 15 min. The final concentration of ACLY was 0.58 μM. Then the solution was aliquoted into 200 μL PCR tubes and heated at the indicated temperature for 3 min. After heating, the samples were centrifuged (12,000 rpm, 10 min, 4 °C) to remove denatured proteins, and the supernatants were saved for analysis.

The SDS–PAGE samples were prepared in 1× loading buffer. A separation gel (8%) and stacking gel (5%) were used to separate and transfer target proteins onto PVDF membranes (0.45 μm). The membranes were blocked at room temperature for 1 h with a blocking solution (Beyotime, Shanghai, China), followed by addition of a primary antibody at room temperature for another 2 h. HRP-linked IgG was used as the secondary antibody and incubated at room temperature for 1 h. Protein bands were imaged by molecular imager ChemiDoc XRS+ (Bio-rad, Hercules, CA, USA).

### 4.6. Analysis of Inhibitory Mechanism of Herbacetin

The inhibition profile of herbacetin against ACLY was determined by the Michaelis–Menten equation and Lineweaver–Burk curve. Reaction rates were recorded separately at different concentrations of substrates with various inhibitor concentrations. Different inhibition types displayed different trends in *K_m_* and *V_max_* (Table 2). For competitive inhibition, *V_max_* is constant, *K_m_* decreases; for noncompetitive inhibition, *K_m_* is constant, *V_max_* decreases; for uncompetitive inhibition, both *V_max_* and *K_m_* decrease, while *K_m_/V_max_* is constant. K_i_ values could be calculated according to the following equations.

### 4.7. Molecular Docking 

Molecular docking was performed using Schrödinger software (LLC, New York, NY, USA, 2021) [35,36]. The co-crystal structure of ACLY was download from the Protein Data Bank (PDB: 6O0H) [20]. The resolution of ACLY crystal structure was 3.67 Å and the co-crystallized ligand was NDI–091143. The pKa value of ACLY was calculated as 7.42 by using PROPKA method with default parameters [37,38]. Protein Prepare Wizard module was used to hydrogenate the complex, remove crystal water and repair loop region [39]. The 3D structure of the protein was minimized using an optimized potentials for liquid simulations-4 (OPLS4) molecular force field. The protein grid box for docking was generated by enclosing the residues in a box with a size of 10 Å × 10 Å × 10 Å centered on the native ligand using the receptor grid generation module with the default settings. The six ligands were docked into the prepared structure by using the Glide module, and the binding energies were scored and ranked by the Glide XP scoring mode.

LigPrep module was used to prepare ligands. The 2D structure of ligands used for docking were created from SMILES obtained from PubChem by CAS number and the 3D structure were generated using LigPrep module. Similarly, the ligand preparation was also performed using the OPLS4 force field. The tautomeric and ionization states were generated between pH 6.5–7.5 by using Epik module [40,41]. The optimized ligands were docked into the prepared ACLY receptor, using Glide XP mode for docking. 

To estimate the free binding energy of ACLY and ligands, and molecular mechanics, the generalized Born model and solvent accessibility (MM–GBSA) analysis was performed using the Prime module of Schrödinger software [42]. The MM–GBSA calculation performed the VSGB solvent model and OPLS4 force field with physics-based modifications for π-π interactions, hydrophobic interactions, and hydrogen bonding self-contact interactions [43,44].

### 4.8. Molecular Dynamics Simulation 

The stability of the interactions within the docked complex constituted by ACLY and the best binding ligand was analyzed by performing molecular dynamic simulation of 200 ns (Schrödinger, LLC, New York, NY, USA, 2021) [42]. The solvent model was SPC, and the atomic framework had been solvated with SPC crystallographic water particles with orthorhombic intermittent limit conditions for a 10 Å buffer region. Na^+^ were added to neutralize the entire framework of atoms. An isothermal isobaric ensemble (NPT) was used to maintain a constant temperature of 300 K and a pressure of 1 bar in the system [45]. A hybrid energy minimization algorithm with 5000 steps of steepest descent followed by conjugate gradient algorithms was used.

### 4.9. Data Analysis

Luminescence signals were processed with Microsoft Excel 2016. IC_50_ and *K_i_* values were calculated by GraphPad Prism 6.0 (GraphPad Software Inc., San Diego, CA, USA). Grayscale values of western blotting bands were analyzed by ImageJ. Numerical results were analyzed by Student’s t test or one-way ANOVA and are presented as the mean ± SD. A *p* value lower than 0.05 was considered significantly different.

## 5. Conclusions

In this work, we identified flavonoids as a new class of ACLY inhibitors with high inhibition potency, expanding the library of ACLY inhibitors. SAR analysis revealed that ortho-dihydroxyphenyl and a conjugated system were critical for activity and inspired high-potency inhibitor design. The selected flavonoid, herbacetin, was highly active and directly bound ACLY with improved thermal stability, indicating that herbacetin is a valuable hit compound. Meanwhile, herbacetin was identified as a noncompetitive ACLY inhibitor, as is illustrated by the docking studies and molecular dynamics simulation. These results deepened our understanding of the interactions between flavonoids and ACLY.

## Figures and Tables

**Figure 1 ijms-23-10747-f001:**
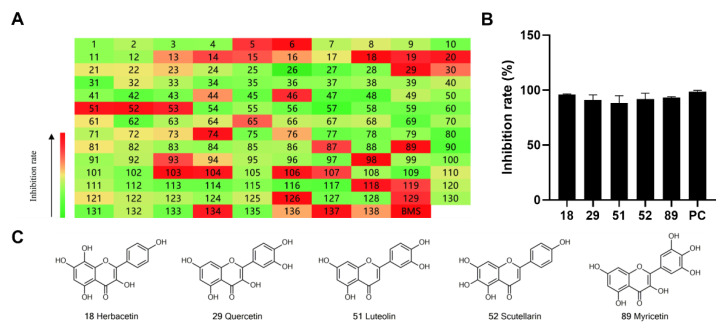
Preliminary screening of 138 natural flavonoids: (**A**) heatmap of the screen results, compounds ID: **1**–**138**; colors from green to red indicate low to high inhibition rates; (**B**) inhibition rates of five ortho-dihydroxyphenyl flavonoids. BMS–303141, positive control (PC); and (**C**) structures of the five flavonoids.

**Figure 2 ijms-23-10747-f002:**
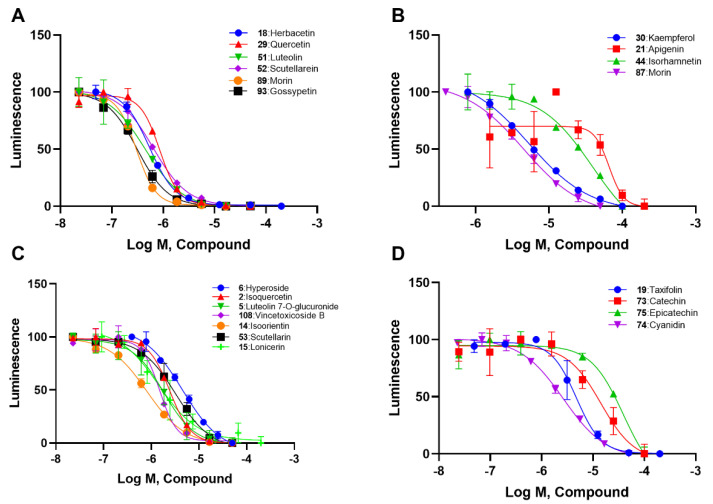
Concentration-response curves of flavonoids in four types: (**A**) ortho–dihydroxyphenyl flavonoids; (**B**) meta–dihydroxyphenyl flavonoids; (**C**) flavonoid glycosides; and (**D**) flavanonols and anthocyanins.

**Figure 3 ijms-23-10747-f003:**
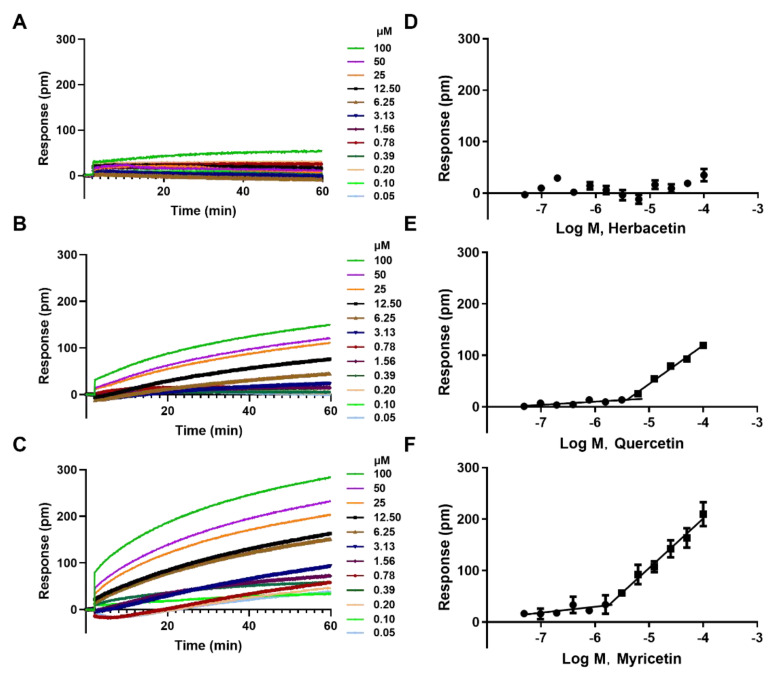
Cell–free aggregation analyses of three typical flavonoids: DMR responses of herbacetin (**A**); quercetin (**B**); and myricetin (**C**); and corresponding concentration–response curves (**D**–**F**).

**Figure 4 ijms-23-10747-f004:**
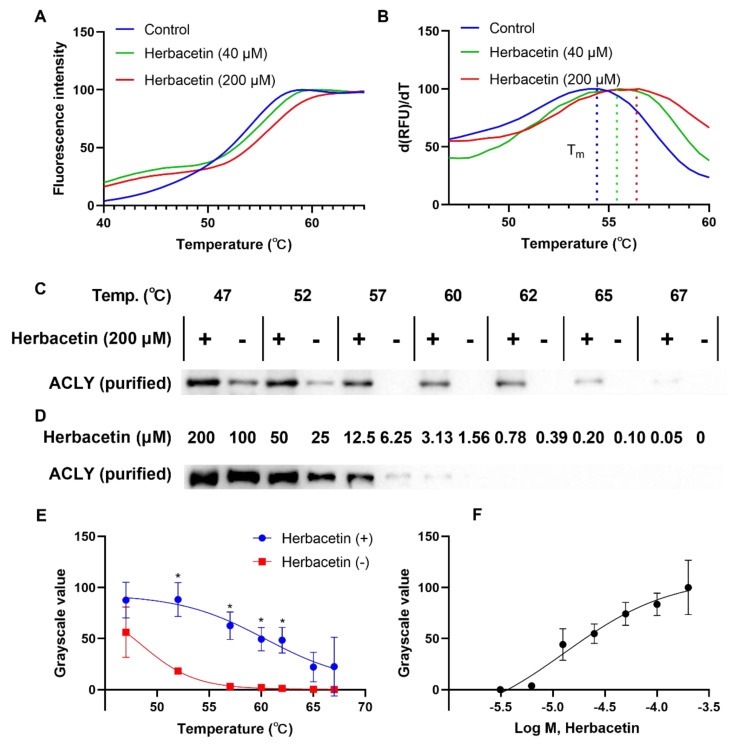
Effects of herbacetin treatment on thermal stability of ACLY: (**A**) melting curve of ACLY in the heating process (fluorescence was recorded by a q–PCR machine); (**B**) first derivative of (**A**), representing the changing rate of fluorescence; (**C**,**E**) Western blot analysis of ACLY on herbacetin treatment compared to DMSO treatment. The statistical results were two independent replicate experiments. * *p* < 0.05; and (**D**,**F**) isothermal dose–response analyses of herbacetin (the temperature was fixed at 57 °C).

**Figure 5 ijms-23-10747-f005:**
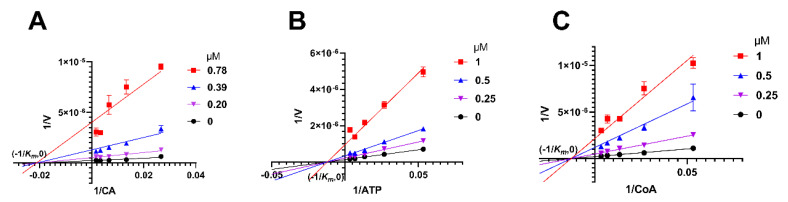
Kinetic studies of herbacetin. Double reciprocal curves were plotted when CA (**A**), ATP (**B**) or CoA (**C**) served as a substrate.

**Figure 6 ijms-23-10747-f006:**
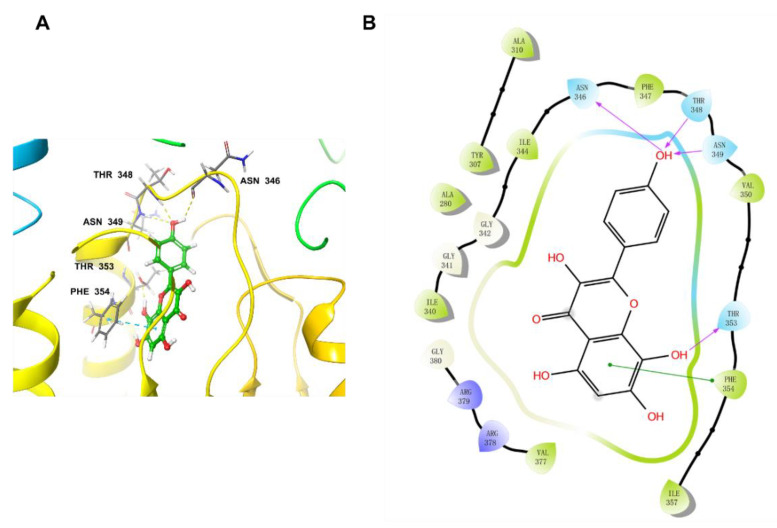
The docking pose of herbacetin binding to ACLY: (**A**) three–dimensional presentation of herbacetin on ACLY. Carbon atoms are shown in green, oxygen atoms are shown in red, and hydrogen atoms are shown in white; hydrogen-bond interactions were showed in dashed lines; and (**B**) two–dimensional presentation of herbacetin and ACLY. Hydrogen-bond interactions were marked with arrows.

**Figure 7 ijms-23-10747-f007:**
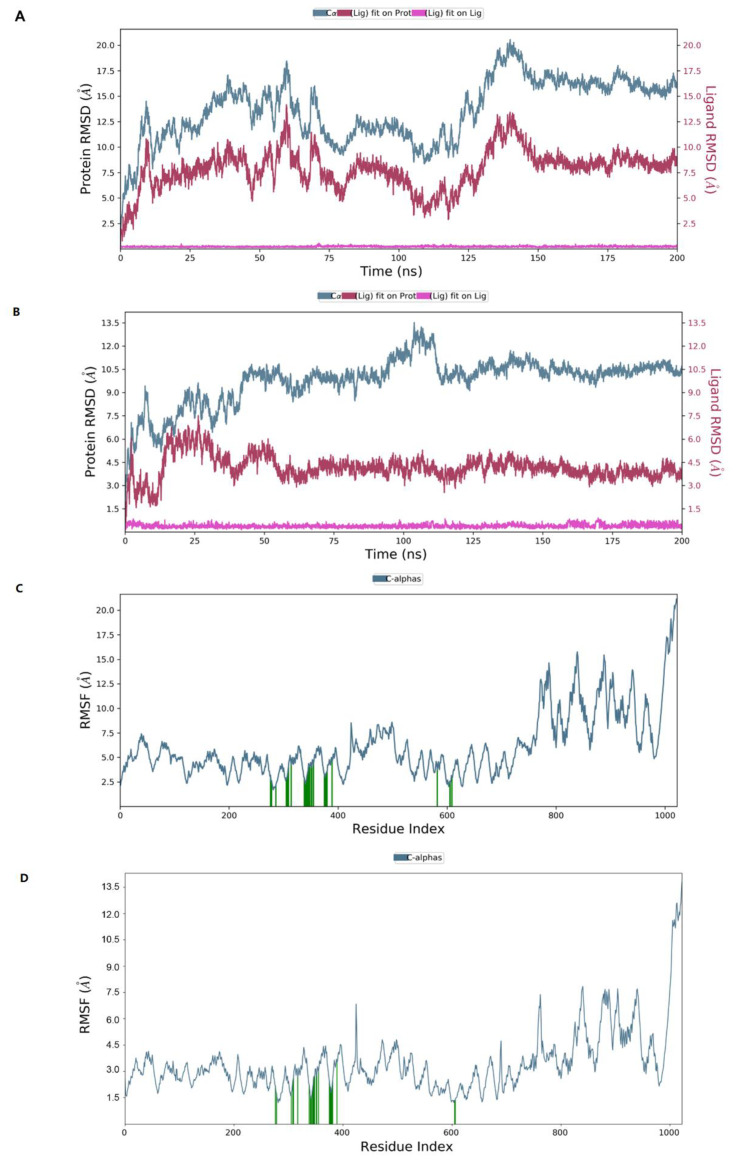
The MD studies of ligands binding to ACLY (**A**); the RMSD plot for ACLY and herbacetin or NDI–091143 (**B**); complex (**C**); the RMSF plot of ACLY chain in the presence of herbacetin or NDI–091143 (**D**).

**Table 1 ijms-23-10747-t001:**
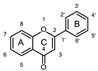
The SAR of the flavonoid inhibiting properties of ACLY.

Entry	Group	Compound	A Ring	B Ring	C Ring	Classification	IC_50_ (μM)
**18**	A	Herbacetin	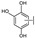	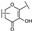	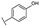	Flavonol	0.50 ± 0.08
**29**	A	Quercetin	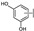	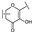	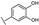	Flavonol	0.86 ± 0.05
**51**	A	Luteolin	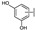	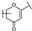	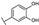	Flavone	0.64 ± 0.13
**52**	A	Scutellarein	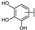	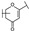	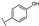	Flavone	0.71 ± 0.06
**89**	A	Myricetin	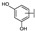	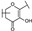	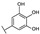	Flavonol	0.57 ± 0.03
**93**	A	Gossypetin	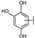	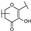	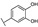	Flavonol	0.31 ± 0.02
**30**	B	Kaempferol	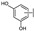	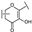	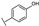	Flavonol	10.16 ± 1.07
**21**	B	Apigenin	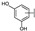	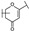	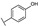	Flavone	19.11 ± 11.63
**44**	B	Isorhamnetin	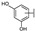	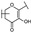	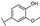	Flavonol	49.74 ± 14.76
**87**	B	Morin	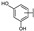	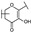	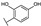	Flavonol	4.07 ± 0.69
**6**	C	Hyperoside	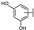	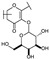	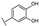	Flavonol	1.53 ± 0.08
**20**	C	Isoquercetin	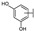	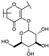	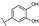	Flavonol	2.14 ± 0.12
**5**	C	Luteolin 7-O-glucuronide	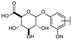	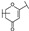	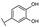	Flavone	1.47 ± 0.20
**108**	C	Vincetoxicoside B	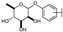	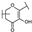	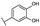	Flavonol	0.57 ± 0.09
**14**	C	Isoorientin	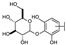	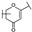	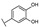	Flavone	1.86 ± 0.29
**53**	C	Scutellarin	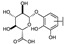	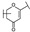	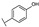	Flavone	3.77 ± 0.26
**15**	C	Lonicerin	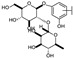	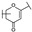	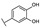	Flavone	1.43 ± 0.44
**19**	D	Taxifolin	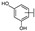	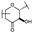	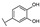	Flavononol	2.54 ± 0.25
**73**	D	Catechin	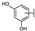	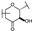	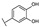	Flavononol	23.64 ± 19.19
**75**	D	Epicatechin	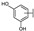	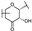	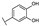	Flavononol	31.63 ± 9.45
**74**	D	Cyanidin	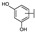	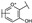	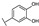	Anthocyanin	1.29 ± 0.14

**Table 2 ijms-23-10747-t002:** Michaelis–Menten equation and constants for different inhibition profiles.

Types	Equations	*V_max_*	*K_m_*
No inhibitor	$v=\frac{V_{max}S}{K_{m}+S}$	*V_max_*	*K_m_*
Competitive	$v=\frac{V_{max}S}{K_{m}\left(1+\frac{I}{K_{i}}\right)+S}$	constant	decrease
Noncompetitive	$v=\frac{V_{max}S}{(K_{m}+S)\left(1+\frac{I}{K_{i}}\right)}$	decrease	constant
Uncompetitive	$v=\frac{V_{max}S}{K_{m}+S\left(1+\frac{I}{K_{i}}\right)}$	decrease	decrease

where *v* is the velocity of the reaction, *V_max_* is the maximum velocity, *S* is the substrate, *K_m_* is the Michaelis constant of ACLY, *I* is the inhibitor and *K_i_* is the inhibition constant. Global shared goodness-of-fit parameters were used to determine the most appropriate inhibition pattern.

## Data Availability

The data supporting this study’s findings are available upon request.

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
