# Peer review of "Discovery of Flavonoids as Novel Inhibitors of ATP Citrate Lyase: Structure–Activity Relationship and Inhibition Profiles"

_ijms, 2022, doi:10.3390/ijms231810747_

Round 1

Reviewer 2 Report

This paper presents high quality research, is well written and presented and therefore qualifies for publication. My only comment is for the presentation of Table 2 to be made clearer, perhaps by using a smaller text font. It would be interesting if the authors could comment on the mechanism on how herbacetin binding to ACLY increases thermal stability. I commend the authors on an excellent paper. 

Reviewer 3 Report

The authors performed an interesting study showing the potential effects of flavonoids (natural compounds) in inhibition of ATP citrare lyase enzyme, which can have beneficial effects in human health by means of the treatment of several health conditons. The study is well planned and presented, but few corrections must be done before its acceptance by IJMS. Regarding the classification of flavonoids presented by the authors, they must re-think their classes (ortho-duhydroxyl, monohydroxil, glycosides, typical flavonoids...) to avoid missunderstanding of the readers. Besides, my other main concern is about the discussion that must be improved, since it is almost only a repetition of the results. In addition, please find attached my detailed comments to the paper. 

Round 2

Reviewer 1 Report

Dear Editor,

The authors have attended all my previous concerns satisfactorily. Then, I can recommend this revised version of manuscript for publication in IJMS.

I just suggest an English revision before publication.

Best regards.